# Amino Acid Residues of the Metal Transporter OsNRAMP5 Responsible for Cadmium Absorption in Rice

**DOI:** 10.3390/plants12244182

**Published:** 2023-12-16

**Authors:** Zhengtong Qu, Hiromi Nakanishi

**Affiliations:** Department of Global Agricultural Sciences, Graduate School of Agricultural and Life Sciences, The University of Tokyo, Tokyo 113-8657, Japan; kyoku@g.ecc.u-tokyo.ac.jp

**Keywords:** OsNRAMP5, cadmium, manganese, rice, transporter, random mutation

## Abstract

The transport of metals such as iron (Fe), manganese (Mn), and cadmium (Cd) in rice is highly related. Although Fe and Mn are essential elements for plant growth, Cd is a toxic element for both plants and humans. OsNRAMP5—a member of the same family as the Fe, Mn, and Cd transporter OsNRAMP1—is responsible for the transport of Mn and Cd from soil in rice. Knockout of *OsNRAMP5* markedly reduces both Cd and Mn absorption, and this *OsNRAMP5* knockout is indispensable for the development of low-Cd rice. However, in low-Mn environments, such plants would exhibit Mn deficiency and suppressed growth. We generated random mutations in OsNRAMP5 via error-prone PCR, and used yeast to screen for the retention of Mn absorption and the inhibition of Cd absorption. The results showed that alanine 512th is the most important amino acid residue for Cd absorption and that its substitution resulted in the absorption of Mn but not Cd.

## 1. Introduction

The global population surpassed 8 billion in 2023 and is increasing; it is projected to exceed 11.2 billion in 2100 [1]. Worldwide, 750 million people suffer from hunger and undernourishment, a number projected to exceed 840 million in 2030 and 2 billion in 2050 [2]. The total global cultivable area has decreased since 1961 as a result of urbanization [3]. The improvements in crop varieties and techniques resulting from the ‘Green Revolution’ have increased yields per unit area; however, further increasing yields is problematic. To provide sufficient food for the increasing global population, there is a need to develop plants that are tolerant of poor environments.

Soil contamination by toxic heavy metals precludes its use for agricultural purposes. When crops absorb nutrients such as trace elements from soil, they also take up harmful heavy metals. Among these harmful heavy metals, cadmium (Cd) is an atypical transition heavy metal readily absorbed in conjunction with other minerals required for plant growth (e.g., iron (Fe), zinc (Zn), and manganese (Mn)) [4]. It has a long biological half-life; high mobility, solubility, fluidity, and bioaccumulation; and long-lasting toxicity, irrespective of concentration [5]. Cd is not essential for plant growth or the biological functions of humans and animals. In plants, excess Cd causes growth disorders. Cd contamination is a severe and ubiquitous environmental problem, and Cd enters food chains by being absorbed by plants and then subsequently accumulating in animals and humans. Lifelong intake of Cd, which has a biological half-life of around 10 years, can damage the lungs, kidneys, bones, and reproductive system. In Japan, Itai-Itai disease was first reported in the 1910s, and Cd-exposed miners in Europe suffered lung damage in the 1930s; in both cases, the damage was induced by chronic Cd intoxication [6]. 

Cd is produced by natural activities (volcanic activity, weathering, and erosion), anthropological activities (smoking, smelting, and fossil fuel combustion), and remobilization of historical sources, including watercourse contamination. Those industrial activities, including mining and smelting, could influence paddy fields to a large extent [7]. As a result, dietary intake accounts for approximately 90% of all Cd intake in the nonsmoking population; other sources include drinking water and exposure to inexpensive jewelry, toys, and plastics [8]. According to the national food survey and estimation of total diet, the Cd intake worldwide is within the range from 0.1 to 0.51 µg/kg of body weight per day, but in comparing the intake of different countries, Asian nations, such as China (0.21–0.51 µg/kg) and Japan (0.31–0.36 µg/kg), showed a higher level of intake than those of the United States (0.13–0.15 µg/kg) and European nations (0.16 µg/kg in Finland, 0.18 µg/kg in Germany, etc.) [9], which could be attributed to the larger consumption of rice in Asian nations [10]. Specifically in China, which is the largest rice producer [11], although the National Standard of the People’s Republic of China limits the Cd content in rice to 0.2 mg/kg [12], 10.3% of rice on the Chinese market exceeds this limit [13]. The independent market surveys carried out in six administrative regions in those three major cropping regions showed tested samples from all administrative regions are Cd-contaminated to different extents: the average Cd content ranged from 0.12 to 0.46 mg/kg and 14–100% exceeded the standard limit [14,15,16,17,18,19]

In rice, Cd is transported within the plant via the apoplastic and symplastic pathways, and both pathways involve transporters of other metallic elements essential for plant growth. Because Cd shares similar chemical properties with Fe, they are closely associated in plants [20]. The mechanisms of the uptake and accumulation of Fe and Cd are somewhat common as a result of similar entry routes within rice. During the vegetative stage, Fe and Cd are absorbed by specific root transporters and transported to the aerial parts via the xylem-to-phloem transfer system, and at grain-filling, grain Fe and Cd are both derived from the phloem [21]. With the presence of Cd, Fe deficiency symptoms could be induced because Cd inhibits not only the absorption of Fe [22], but the transportation of Fe from root to shoot [23]. On the other hand, the addition of Fe could also reduce Cd content in rice [20] and enhance rice growth and yield [24], which suggests that Cd translocation into rice might occur via Fe metabolic pathways [25]. The interaction between Mn and Cd has also been identified, because the accumulation of Cd is reduced in both roots and shoots in the Mn sufficiency environment compared with the Mn deficiency environment [26]. Fe and Mn alleviated Cd toxicity by preventing Cd from being absorbed by forming an Fe plaque on the surface of rice roots [27]. Meanwhile, Fe and Mn could also protect plants from damage induced by Cd on root growth and photosynthesis [28].

Several genes in rice have been reported to take part in xylem loading and phloem redistribution of Fe, Mn, and Cd at different locations in the plants [29]. For example, members of the heavy metal-associated protein (HMA) metal-transporter family transport Cd to the root vascular bundle. Similar to AtHMA4 and AhHMA4, OsHMA2 has also been identified as a transporter of both Zn and Cd, and in OsHMA2-suppressed rice, the concentrations of both Cd and Zn decreased in the leaves and seeds, which suggests that OsHMA2 plays a role in Cd loading to the xylem and participates in root-to-shoot translocation of Cd apart from Zn [7]. Different from OsHMA2, OsHMA3 reportedly does not transport other metals such as Zn [30]. To be specific, OsHMA3, a regulator for Cd transport in the xylem in rice, has the function of mediating vacuolar sequestration of Cd in root cells [31]. The expression of *OsHMA3* was directly proportional to Cd concentration in the environment [32], but with excessive Fe treatment, the expression of *OsHMA3* significantly increased [33]. RNAi-mediated knockdown of *OsHMA3* increased root-to-shoot Cd translocation, and on the other hand, the overexpression of *OsHMA3* reduced shoot Cd accumulation, which indicates that OsHMA3 has the function in vacuolar compartmentation of Cd in roots, which decreases the xylem loading of Cd and subsequent shoot Cd accumulation [34]. Cd is also transported to seeds via the phloem in a manner involving the product of *OsLCT*; the phloem and seeds of *OsLCT1* mutants generated through RNA interference had low levels of Cd [35]. Because Cd is toxic, it is detoxified by inclusion in complexes with thiol compounds such as phytochelatin (PC) and glutathione (GS, a synthetic substrate for PC). In rice, such thiol compounds are synthesized by OsGS and OsPCS, resulting in the extracellular transport of some Cd [36]. Therefore, it is necessary to modify steps in the plant Cd transport pathway—for instance, Cd absorption from soil, transportation from root to leaf, and sequestration into the vacuole—to enhance its detoxification. Doing so would enable the development of low-Cd foods in which Cd is not stored in seeds.

The natural resistance-associated macrophage protein (NRAMP) family is involved in the absorption of metal elements in diverse taxa. NRAMP1 transports divalent metals (e.g., Mn, Fe, and cobalt) across the phagosomal membranes of macrophages, as does divalent metal transporter 1 (DMT1; alternatively, NRAMP2, DCT), which is a transporter of Cd and Fe [37]. The NRAMP family serves as the secondary active transporters with the general features of proton transportation and proton-metal coupling, and the alternating access in the NRAMP family depends largely on the motion and the structure of transmembrane proteins [38]. Rice has seven NRAMP transporters, among which OsNRAMP1 is responsible for the uptake and transport of Cd in plants [39]. Transformation with OsNRAMP1 reduced the Cd tolerance of yeast [40]. However, OsNARMP1 also transports Mn and Fe. Similar interactions between Cd and Fe were also found in both the ferrous Fe transporter iron-regulated transporter 1 (IRT1) and IRT2 in rice. Both OsIRT1 and OsIRT2 are related to Fe uptake in roots and also showed an influx activity of Cd as well as Fe in yeast, showing that OsIRT1 and OsIRT2 are important transporters in roots with the function of the uptake of Cd [41,42]. OsIRTs may contribute to the uptake of Cd in aerobic conditions when water is released. Meanwhile, Cd is absorbed in roots through the OsNRAMP5 transporter, and OsNRAMP5, which has been identified as a transporter of Mn and Cd, is responsible for the absorption of Mn and Cd from soil [43]. The reason that rice accumulates more Cd than other cereal crops may also be related to the *OsNRAMP5* gene having a higher expression in rice [29]. Interestingly, Fe absorption by OsNRAMP5 in root and shoot tissues did not differ significantly between the wild type and an *OsNRAMP5* mutant [44]. Furthermore, knockout of *OsNRAMP5* markedly reduced the amount of Cd in rice by abolishing its uptake from the soil. Therefore, knockout of OsNRAMP5 is a promising trait for producing low-Cd rice. Because OsNRAMP5 transports both Mn and Cd, *OsNRAMP5* knockout also reduced Mn absorption by about 90% [45]; therefore, in low-Mn environments, such plants would exhibit Mn deficiency and suppressed growth. 

Mutations in OsIRT1 and AtNRAMP4 alter their metal selectivity [46,47]. Furthermore, the changes in structure of ScaNRAMP also vary the metal transportation [48], which might result from a single amino acid substitution together with the protein stability [49]. Similarly, among the 538 amino acid residues comprising OsNRAMP5, 1 or more may mediate its transport of Mn, Cd, or both. Therefore, substitution of a specific amino acid residue may affect Mn and/or Cd transport in a manner that does not alter the Mn uptake while suppressing Cd uptake by changing the amino acid, the protein structure, or both. Rice with such a mutation could maintain Mn uptake while avoiding Cd accumulation when grown in Cd-contaminated soil with low Mn, with no negative influence on the growth. The development of rice varieties that can absorb Mn but not Cd would enable the cultivation of soil with a greater range of Cd contamination levels than would rice varieties with *OsNRAMP5* knockout, and enlarge the production of sufficient crops with low Cd concentration. To this end, in the present study, we introduced mutations into OsNRAMP5 and evaluated their effects on Mn and Cd transport.

## 2. Results

### 2.1. Optimization of Mn and Cd Concentration for Mutant Screening in Yeast

The appropriate conditions of screening were determined by analyzing the transportation of Cd and Mn by OsNRAMP5 in yeast because although when expressed in yeast, OsNRAMP5 functions as a transporter of both Mn and Cd, the growth of the yeast might also depend on the environmental concentrations of the metals [42,44]. In the absence of Mn with different concentrations of EGTA, the growth of the negative control (VC) was inhibited from 10 mM EGTA, and the effect was the most at 20 mM EGTA (Figure 1b,c), but no significant difference was found with 2 mM EGTA (Figure 1a); however, OsNRAMP5-expressing N5 showed good growth even with 20 mM EGTA as a result of transport of Mn by the OsNRAMP5 (Figure 1a–c). In the presence of different concentrations of Cd, the growth of N5 was impaired from 50 μM Cd, and the most significant difference was found at 100 μM Cd (Figure 1e,f), but the Cd concentration of 10 μM did not influence the growth of the yeast significantly (Figure 1d), and that of VC was unaffected regardless of Cd concentration (Figure 1d–e). Furthermore, both VC and N5 showed inhibited growth in a −Mn/+Cd environment compared with a +Cd environment and −Mn environment, respectively (Figure 1c,e,f,g), and no significant difference was found between the growth of VC and N5 in a −Mn/+Cd environment (Figure 1g). These results indicate that the difference in growth of VC and N5 could be clearly identified in the environment with 20 mM EGTA chelating Mn and 100 μM CdCl_2_, which demonstrates that the concentrations of EGTA and Cd were appropriate for the later screening.

### 2.2. Patterns of OsNRAMP5 Mutations

We selected 100 colonies from 10 −Mn/+Cd plates in the first screening (200,000 colonies in total) and 20 colonies in the second screening (Table 1). The nucleotide sequences of 20 mutants were classified into 4 patterns: pattern 1, 2 nucleotide mutations corresponding to 1 amino acid mutation (A512T and a silent mutation at the 21st position); pattern 2, 4 nucleotide mutations corresponding to 3 amino acid mutations (S8R, C111Y, A512T, and a silent mutation at the 291st position); pattern 3, 2 nucleotide mutations corresponding to 1 amino acid mutation (A512T and a silent mutation at the 507th position); and pattern 4, 3 nucleotide mutations corresponding to 3 amino acid mutations (S8R, C111Y, and A512T). Only mutants with different amino acids were focused on in this study, so these four patterns were divided into type 1 and type 2 according to their amino acid mutations: patterns 1 and 3 with A512T and patterns 2 and 4 with S8R, C111Y, and A512T. The G to A substitution at position 1534 (alanine to threonine at residue 512) was present in all four patterns.

### 2.3. Mutants Absorb Mn but Not Cd

The absorption of Cd and Mn by the 20 mutants was compared with those of the VC and N5. Under Mn-deficient conditions, all mutants showed growth similar to N5, indicative of similar levels of Mn absorption (Figure 2a). In the presence of Cd, N5 showed little growth, but the mutants—particularly those with pattern 3 (M35, M36, M70, and M82)—showed improved growth compared with the other mutants (Figure 2b). In the presence of Cd but not Mn, the growth of VC and N5 was inhibited by Mn deficiency and Cd toxicity, respectively. However, all mutants showed improved growth, indicating that they could absorb Mn but not Cd in the environment of −Mn/+Cd (Figure 2c).

The patterns with the biggest number of plasmids were selected from each type (pattern 1 from type 1 and pattern 2 from type 2) to test the sensitivity to EGTA and Cd. M6 (mutant from pattern 1) and M9 (mutant from pattern 2) were investigated on media with various concentrations of EGTA and Cd. Apart from the result of VC and N5 similarly to Figure 1, because VC was not sensitive to low concentrations of Mn (2 mM EGTA), the mutants showed only slightly better growth with 10 mM EGTA and a significant difference with 20 mM EGTA (Figure 3a–c). Meanwhile, the growth of the mutant was slightly inhibited with 10 mM of EGTA compared with N5, but when N5 also showed decreased growth with a further increase in EGTA, the mutants had growth similar to it (Figure 3b,c). In the presence of Cd, the mutants were not inhibited in growth as N5 at 10µM Cd (Figure 3d), and even showed slightly better growth at medium concentration (50 μM) compared with VC (Figure 3e). With a high concentration of Cd (100 μM), the growth of the mutants was slightly worse than that of VC (Figure 3f). Meanwhile, the significant difference between the growth of M6 and M9 was not found in either −Mn or +Cd conditions (Figure 3).

In the growth test in liquid medium, the growth rates of M6 and M9 were compared with those of VC and N5. Under Mn-deficient conditions, M6 and M9 grew slightly slower than N5, but much faster than VC (Figure 4a). In the presence of Cd, the growth of N5 was suppressed, and the growth of the two mutants was similar to that of VC (Figure 4b). In the presence of Cd but not Mn, the growth rates of the two mutants were higher than those of VC and N5 (Figure 4c).

The growth rate of VC, N5, M6, and M9 in different concentrations of Cd for 24 h indicated that both mutants were more tolerant in all Cd concentrations compared with the N5 yeast, but similar in growth compared with the VC yeast. Both mutants showed similar growth rates regardless of the concentration of Cd, but N5 showed a decreased growth rate at high Cd concentration (100 μM CdCl_2_) compared with low Cd concentration (10 μM CdCl_2_) (Figure 4d).

### 2.4. Mutants Show Reduced Absorption of Cd but Similar Absorption of Mn Compared with N5

M6 and M9 showed Cd absorption similar to that of VC but significantly different from that of N5 in the presence of 20 μM CdCl_2_ for 72 h (Figure 5a). In the presence of Cd, M6, M9, VC, and N5 showed similar Mn absorption. In the absence of Cd, M6 showed greater Mn absorption than N5, and M9 showed Mn absorption lower than N5 but similar to VC (Figure 5b). 

M6 showed similar Fe absorption to N5 in the absence of Cd, which was significantly higher than in the presence of Cd. However, M9 showed similar absorption of Fe to VC irrespective of the presence of Cd. Meanwhile, the absorption of Fe in both M9 and VC showed lower levels in the absence of Cd, but M6, M9, VC, and N5 showed similar levels of Fe absorption in the presence of Cd (Figure 5c). There was no significant difference in the absorption of Cu compared with M6, M9, VC, and N5, irrespective of the presence of Cd. Therefore, neither pattern 1 mutants nor the pattern 2 mutants showed an influence in the absorption of Cu. Moreover, M6, M9, VC, and N5 showed similar Zn absorption in the presence of Cd, although in the absence of Cd, only M6 showed elevated Zn absorption (Figure 5e).

### 2.5. Alanine 512 Is Essential for Cd Absorption

The 20 plasmids harbored a substitution of alanine for threonine at residue 512. To evaluate its importance, we mutated alanine 512 to methionine (A512M), isoleucine (A512I), and aspartic acid (A512D). The metal-transport activities of the mutants were compared with those of VC, N5, and A512T. All mutants had growth rates similar to A512T (Figure 6).

## 3. Discussion

The growth of the mutants in the presence of Cd and the absence of Mn is likely attributable to the mutations (Table 1 and Figure 2a,b, Figure 3 and Figure 4a,b). In the medium containing Cd but not Mn, VC and N5 did not grow, as a result of Mn deficiency and Cd toxicity, respectively (Figure 1 and Figure 2), and in the environment with the presence of Cd and the absence of Mn, the improved growth of N5 compared with that in the environment with the presence of Cd (Figure 1f,g) might be attributed to the chelation by EGTA of Cd, which reduced the free Cd in the medium [50]. However, the obtained *OsNRAMP5* mutants showed good growth (Figure 2). The selected M6 and M9 were investigated and the transport of Mn remained, but they had a low sensitivity to Cd (Figure 3). In liquid medium, the growth rates of the M6 and M9 mutants were similar to that of N5 in the absence of Mn (Figure 4a); both mutants showed similar growth rates under all conditions (Figure 4a–c) and a lower Cd sensitivity in different Cd concentrations compared with N5 (Figure 4d). These findings indicate that M6 and M9 had reduced absorption of Cd in various concentrations of Cd (Figure 2b,c, Figure 3d–f, Figure 4c and Figure 5a) but similar or increased absorption of Mn compared with N5 (Figure 2a,c, Figure 3a–c and Figure 5b), suggesting that the *OsNRAMP5* mutants could mediate the absorption of Mn while suppressing that of Cd. Also, alanine 512, which was common to all mutations, is likely involved in the absorption of Cd (Figure 6). However, even though there was a significant increase in Mn absorption level of M6 and N5 in the absence of Cd, it still showed similar performance with the presence of Cd in M6, M9, VC, and N5 (Figure 5b) due to competitivity of Mn with Cd [51]. Furthermore, because the previous research on OsIRTs indicated that although Cd accumulation in the roots and shoots of *OsIRT1*-overexpression plants was increased under MS medium with excessive Cd, such a phenotype was not shown in the paddy field, which demonstrates that the contribution of the transporters is also affected by the external environmental conditions [52]. The mutations affecting the Mn and Cd transport ability of yeast will be introduced into rice to verify whether rice with mutations can show a higher Mn concentration in the environment with low Mn and lower Cd in the environment with Cd, which is similar to the performance of yeast in this study, in the future.

In the presence of Cd, M6 showed lower Cd absorption, and in the absence of Cd, M6 showed higher Mn (Figure 5a,b), suggesting that the altered Cd and Mn absorption of M6 resulted from the mutation of alanine 512 (Figure 6), and the similar absorption level of Mn in M9 compared to VC might be attributed to the extra mutation of serine 8 and/or cysteine 111 (Table 1 and Figure 5b). Alanine 512 is also important for the absorption of Zn. In the absence of Cd, Zn absorption by N5 was markedly lower than that of M6, and slightly but non-significantly lower than that of M9 (Figure 5e). However, because Cd is more competitive and more easily absorbed than Zn [51], the Zn absorption by M6 and M9 decreased to a level similar to that of VC and N5 in the presence of Cd. Cd is less competitive with Cu [51], which might lead to no significant change in the absorption of Cu with both the presence and absence of Cd in all M6, M9, VC, and N5 (Figure 5d). Moreover, mutations of serine 8, cysteine 111, or both may impede the absorption of Zn and Fe, possibly explaining the similar Zn absorption of N5 and M9 and the lower Fe absorption in M9 than in N5 and M6 in the absence of Cd (Figure 5c,e). For all metals investigated, M9 had similar absorption rates to VC (Figure 7), suggesting that serine 8, cysteine 111, or both are important for metal transport by OsNRAMP5. In M9, the A512T mutation non-significantly enhanced Mn transport compared with VC (Figure 5b). This may explain why the mutations of patterns 2 and 4 were obtained by screening in the absence of Mn.

Particular attention should be paid to the change at nucleotide 507, because yeast with pattern 3 grew better than that with pattern 1 on a Cd-containing medium with the same construction of amino acid (Table 1 and Figure 2b). Meanwhile, the change at nucleotide 21 in M6 might also be important to enhance the absorption of Mn and Zn in the absence of Cd, because M6 showed an increased absorption of Mn and Zn compared with N5, but M9 sharing the mutation at alanine 512 showed a reduced and similar absorption of Mn and Zn, respectively, compared with N5 (Figure 5b,e). The changes from C to T at nucleotide 21 (pattern 1) and T to C at nucleotide 507 (pattern 3) (Table 1) could alter transcriptional efficiency, RNA stability, transfer RNA levels, and protein expression levels even though the amino acid was kept the same [53,54]. It will be very interesting to see if these silent mutations have the same effect in plants as well as in yeast. Whether these are silent mutations that affect absorption could be evaluated by creating a plasmid with the only mutation at nucleotide 21 and a plasmid with the only mutation at nucleotide 507 and evaluating the effect on metal absorption compared with M6, A512T, and yeast with the pattern 3 mutant introduced into both yeasts and plants in a future study. Because all three mutations were not found in other NRAMP proteins, we need further study on whether the influence of these residues is also conserved in other NRAMPs.

Regarding structural prediction, OsNRAMP5-Q337K, in which a glutamine residue in the eighth transmembrane domain was substituted with a lysine residue, reduced the grain Cd concentration without causing severe Mn deficiency in rice [55]. The three mutations in this study were predicted to be cytoplasmic, and all three mutations slightly changed the structure of the protein (Figure 7). It is necessary to investigate how these residues, which are not extracellular and so cannot interact directly with extracellular metal ions, affect the absorption of metals.

## 4. Materials and Methods

### 4.1. Mn and Cd Absorption Assays

The plasmids pDR195 and pDR195 containing *OsNRAMP5* were transformed into the Mn-absorbing transporter-deficient mutant yeast strain Δ*smf1* (MATalpha, his3Δ1; leu2Δ0; meta15Δ0; ura3Δ0; YOL122c:; kanMX4), generating VC and N5, respectively. Metal transport assays were carried out in synthetic defined (SD) medium (2% glucose, 0.5% yeast nitrate base without amino acids, and 2% agar) containing Cd (10 μM, 50 μM, 100 μM CdCl_2_) but not Mn (2 mM, 10 mM, 20 mM EGTA, pH 5.9) in different concentrations [43,44]. The media were spotted with 8 µL of yeast suspension (OD_600_ = 0.1, 0.01, 0.001), incubated statically at 30 °C for 2 days, and the growth of the yeast strains was monitored.

### 4.2. Error-Prone PCR

To ligate *OsNRAMP5* into pDR195, *Hind*III and *EcoR*I sites were introduced (Ligation Mighty Mix (TaKaRa)), and the vector was digested with *Bam*HI and *Xhol*I. Mutations were introduced into *OsNRAMP5* via error-prone PCR, which we employed previously [56], in ten separated tubes (using 50 × Titanium Taq DNA Polymerase (TaKaRa)). The PCR conditions were denaturation at 95 °C for 15s, annealing at 55 °C for 15 s, and extension at 68 °C for 2 min for 30 cycles, during which the Mn concentration was changed and random mutations were introduced. OsNRAMP5 has 1614 nucleotides; we used 300 μM Mn to introduce two to five mutations.

### 4.3. Screening

OsNRAMP5 fragments with random mutations obtained from 10 separated tubes in the error-prone PCR were ligated into pDR195. The vectors with OsNRAMP5 fragments from each error-prone PCR tube were transformed into Δsmf1 and cultured on −Mn/+Cd SD medium (20 mM EGTA, 100 μM CdCl_2_, pH = 5.9) for 48 h at 30 °C in separated plates for the first screening (10 plates). Next, colonies were diluted and cultured in fresh −Mn/+Cd SD medium for the second screening (100 plates) and the colonies were sequenced.

### 4.4. Growth Assay

Plasmids harboring *OsNRAMP5* mutants were transformed into Δ*smf1* and cultured on solid −Mn, +Cd, and −Mn/+Cd SD media. The EGTA and Cd sensitivity of the mutants was tested on the solid +Cd media with different conditions (2 mM, 10 mM, 20 mM EGTA and 10 μM, 50 μM, 100 μM CdCl_2_) using the same method as 4.1. The growth rates of the mutants were analyzed in liquid −Mn, +Cd, and −Mn/+Cd SD media starting with OD_600_ = 0.05 and compared with those of VC and N5 within 48 h. The Cd sensitivity was also analyzed by measuring the growth rate of the mutants in +Cd SD media with different concentrations of Cd and comparing them with those of VC and N5 in 24 h.

### 4.5. Uptake of Metals

VC, N5, and two *OsNRAMP5* mutant yeast strains (M6 and M9) were cultured in 50 mL of liquid SD medium (20 µM CdCl_2_, 2.5 μM MnSO_4_, 0.75 μM FeCl_3_, 1.5 μM ZnSO_4_, 0.15 μM CuSO_4_) for 72 h at 30 °C. The yeasts were centrifuged at 8000× *g* for 5 min, washed with 50 mL of distilled H_2_O at 8000× *g* for 5 min 2 times, and washed with 30 mL of distilled H_2_O at 8000× *g* for 5 min. The pellets were dried in a warmer at 60 °C for 48 h. The dried pellets were weighed and wet-ashed diluted with 2% HNO_3_, and the Fe, Mn, Cu, Zn, and Cd concentrations were analyzed with inductively coupled plasma optical emission spectrometry (ISPS-3500, Seiko Instruments Inc., Chiba, Japan).

### 4.6. Amino Acid Substitution

The candidate amino acid residue in pDR195 (digested by *Sal*I) was substituted for other amino acid residues to evaluate its function in Mn and Cd transport. The resulting plasmids were transformed into Δ*smf1* and cultured on solid −Mn, +Cd, and −Mn/+Cd SD media.

### 4.7. Statistical Analysis 

The significance of the differences was evaluated using one-way analysis of variance followed by Tukey’s multiple comparison test; *p* < 0.05 was considered to indicate significance.

### 4.8. Protein Structure Analysis

The schematic diagram of the OsNRAMP5 protein structure was produced with AlphaFold DB version.

## 5. Conclusions

The results in this study indicate that alanine 512 mediates, at least in part, Cd transport by OsNRAMP5; its substitution significantly decreased Cd transport but increased Mn transport. Furthermore, it is also necessary to consider the two amino acid residue changes, S8R and C111Y, other than the 512th alanine for the metal selectivity of the OsNRAMP5 protein. Rice harboring metal-selective OsNRAMP5 could grow in low-Mn soil and/or Cd-contaminated soil, thereby increasing the cultivable land area in the future.

## Figures and Tables

**Figure 1 plants-12-04182-f001:**
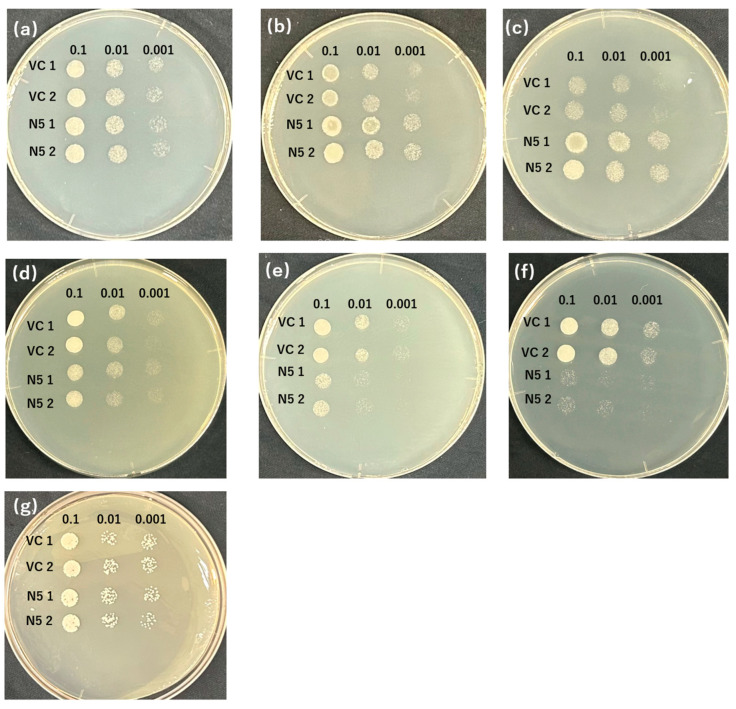
OsNRAMP5 transport activity in yeast. Growth of two individual manganese-absorbing transporter-deficient mutant Δ*smf1*-harboring empty pDR195 vectors (VC 1 and 2) and *OsNRAMP5* (N5 1 and 2) in synthetic defined medium containing (**a**) 2 mM EGTA, (**b**) 10 mM EGTA, (**c**) 20 mM EGTA, (**d**) 10 μM CdCl_2_, (**e**) 50 μM CdCl_2_, (**f**) 100 μM CdCl_2_, (**g**) 20 mM EGTA, and 100 μM CdCl_2_.

**Figure 2 plants-12-04182-f002:**
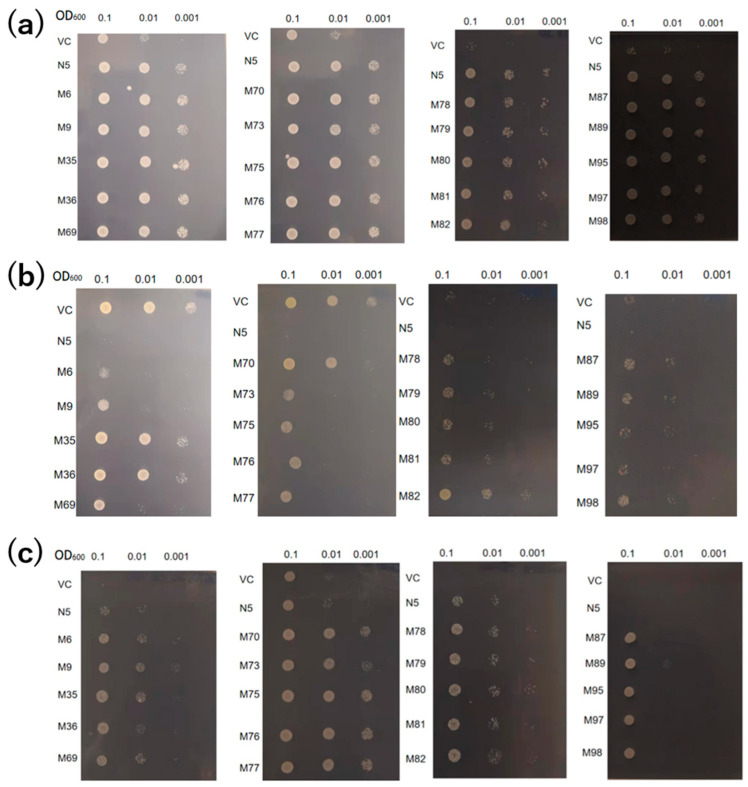
Transport activities of *OsNRAMP5* mutants in Δ*smf1* yeast expressing the *OsNRAMP5* mutants (M), empty pDR195 vector (VC), and wild-type *OsNRAMP5* (N5) in solid synthetic defined medium containing (**a**) 20 mM EGTA, (**b**) 100 μM CdCl_2_, and (**c**) 20 mM EGTA and 100 μM CdCl_2_.

**Figure 3 plants-12-04182-f003:**
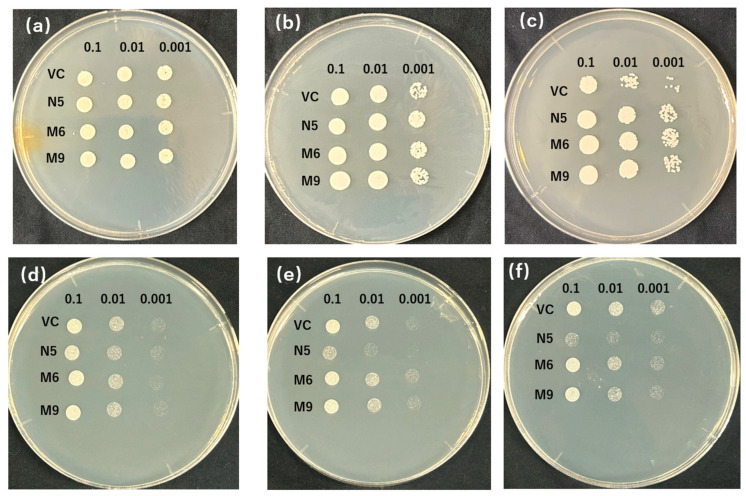
EGTA and Cd sensitivity of *OsNRAMP5* mutants in Δ*smf1* yeast expressing the *OsNRAMP5* mutants (M), empty pDR195 vector (VC), and wild-type *OsNRAMP5* (N5) in solid synthetic defined medium containing (**a**) 2 Mm EGTA, (**b**) 10 Mm EGTA, (**c**) 20 Mm EGTA, (**d**) 10 μM CdCl_2_, (**e**) 50 μM CdCl_2_, and (**f**) 100 μM CdCl_2_.

**Figure 4 plants-12-04182-f004:**
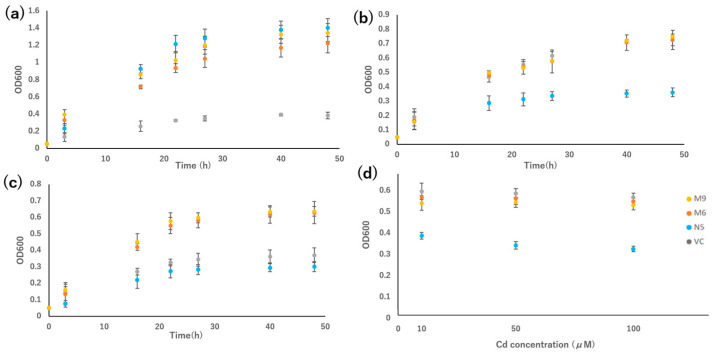
Growth rate of *OsNRAMP5* mutants in Δ*smf1* yeast expressing the *OsNRAMP5* mutants (M6 and M9), empty pDR195 vector (VC), and wild-type *OsNRAMP5* (N5) in liquid synthetic defined medium containing (**a**) 20 mM EGTA, (**b**) 100 μM CdCl_2_, (**c**) 20 mM EGTA and 100 μM CdCl_2_, and (**d**) 10 μM, 50 μM, and 100 μM CdCl_2_.

**Figure 5 plants-12-04182-f005:**
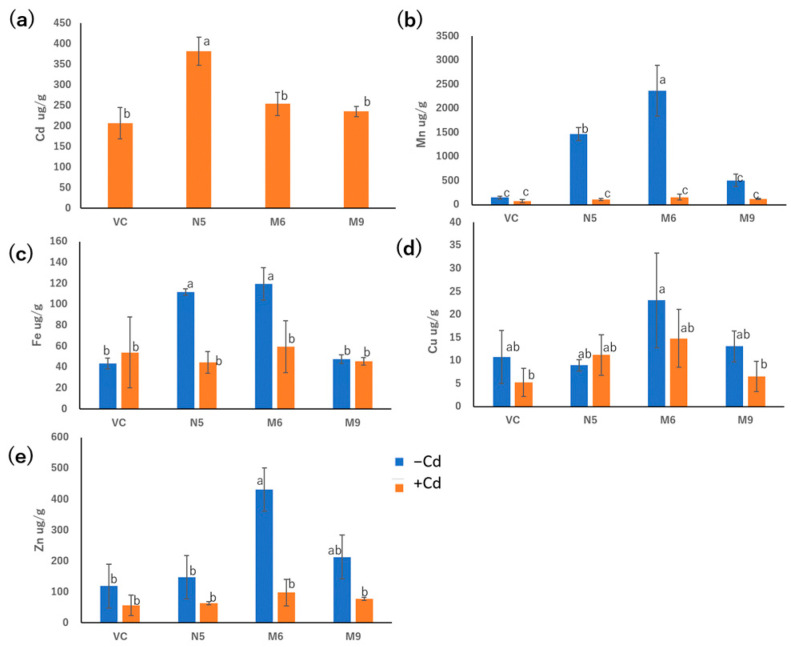
Metal concentrations in empty pDR195 vector (VC), wild-type OsNRAMP5 (N5), and mutants (M6 and M9) in the presence or absence of 20 μM CdCl_2_ for 72 h followed by drying for 24 h: (**a**) cadmium, (**b**) manganese, (**c**) iron, (**d**) copper, and (**e**) zinc. Data are the means ± standard deviation (*n* = 3). Different letters indicate significant differences (*p* < 0.05).

**Figure 6 plants-12-04182-f006:**
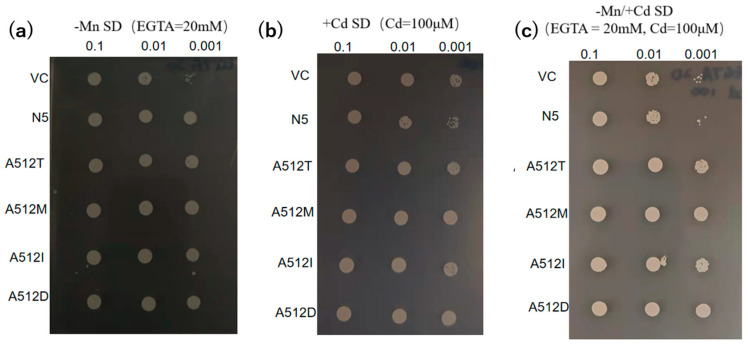
Transport activities of *OsNRAMP5* mutants. Growth of empty pDR195 vector (VC), wild-type *OsNRAMP5* (N5), and amino-acid-512 mutants in synthetic defined medium containing (**a**) 20 mM EGTA, (**b**) 100 μM CdCl_2_, and (**c**) 20 mM EGTA and 100 μM CdCl_2_.

**Figure 7 plants-12-04182-f007:**
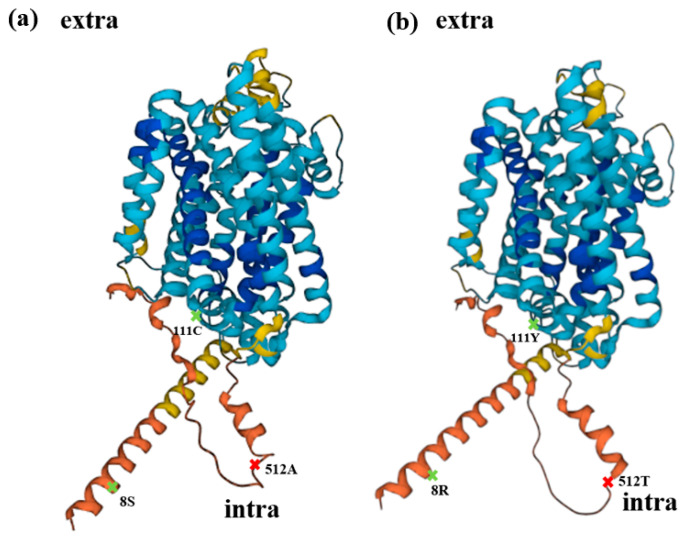
Structural prediction. (**a**) Schematic diagram of wild-type OsNRAMP5 produced with AlphaFold DB version, with alanine 512 (red), which is important for cadmium and manganese transport and affecting the absorption of zinc, and serine 8 and cysteine 111 (green), which influence metal absorption. (**b**) Schematic diagram of OsNRAMP5 with three mutations predicted with AlphaFold DB version, with potentially important locations indicated by the same color as (**a**).

**Table 1 plants-12-04182-t001:** Sequences of mutant *OsNRAMP5* and the locations of the mutations from the start codon of *OsNRAMP5*.

Pattern	Plasmid No.	Mutation Location	Change in Base	Change in a.a.
1	6, 69, 75, 76, 77, 78, 79, 80, 81, 87, 95, 97, 98	#21	C→T	−
		#1534	G→A	A→T
2	9, 73	#22	A→C	S→R
		#291	T→C	−
		#332	G→A	C→Y
		#1534	G→A	A→T
3	35, 36, 70, 82	#507	T→C	−
		#1534	G→A	A→T
4	89	#22	A→C	S→R
		#332	G→A	C→Y
		#1534	G→A	A→T

## Data Availability

All data are contained in the article.

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
