# Peer review of "Amino Acid Residues of the Metal Transporter OsNRAMP5 Responsible for Cadmium Absorption in Rice"

_plants, 2023, doi:10.3390/plants12244182_

Round 1

Reviewer 1 Report

Comments and Suggestions for Authors

Qu and Nakanishi have performed a screen to identify amino acid residues involved in cadmium (Cd) transport by the metal transporter OsNRAMP5.They have mutagenized the cDNA encoding OsNRAMP5 using error prone PCR and selected for mutations that confer lower sensitivity to toxic concentration of Cd. They show that some of the mutations they obtain decrease Cd accumulation in yeast.

The work is relevant because OsNRAMP5 is the main pathway for Cd absorption in rice. AsNRAMP5 is also important for manganese nutrition in this species. The authors have obtained promising results, but the study is still too preliminary, and many additional experiments are required to reach robust conclusions. The result section lacks important information about the experiments. The introduction and the discussion lack reference and comparison to other work on NRAMP transport selectivity.

Major comments

1) The introduction is solely focused on the mechanisms of Cd accumulation in rice and it lacks information on the mechanisms of metal transport by NRAMP proteins, structural information on NRAMP proteins and previous work on the structure-function relationships in this protein family. Especially, previous work already identified mutations that decrease Cd transport by other NRAMPs based on structural information (Bozzi et al., 2016) or on screen similar to the one that the authors report (Pottier et al., 2014).

2) in figure 1, the authors show that, when expressed in yeast, OsNRAMP5 is able to complement the growth defect of the smf1 mutant on medium supplemented by EGTA and confers increased sensitivity to Cd. Both facts are firmly established in the literature and do not need to be illustrated again. In contrast, the authors do not show what happens in the conditions they have used for screening.

As the reader has to guess, the screen combines EGTA and Cd supplementation The authors have not considered the fact that EGTA binds Cd. Therefore, the free Cd concentration in the screening conditions is probably much lower than in the treatment with Cd only.

2) Line 151-159: as mentioned above, the conditions of the screen and the library that was used for screening are not described. Based on the results shown in table 1, the same plasmids have been recovered many times. It is thus difficult to infer how many independent clones and mutations have been screened. Whether the screen was saturated is not indicated or discussed in the manuscript.

3) Line 163: the authors have not studied “Cd or Mn absorption” by the 20 mutants, as they claim. They have merely confirmed, looking at yeast growth, that they are still able to transport manganese and display lower sensitivity to Cd. This should be corrected.

Similarly, line 210, the authors have not measured any “metal-transport activity”, as they claim but again used yeast growth as a proxy to ability of the transport ability. This should be corrected.

4) All the mutants identified have more than one mutation. To conclude about the effect of individual mutations, the authors have to generate single mutants using site-directed mutagenesis. This was done for the A512T (fig 5) but should also be done for the 2 other mutations reported.

5) The authors have confirmed the effect of the mutations only on one concentration of Cd. To accurately determine the change in Cd sensitivity, different Cd concentrations should be tested.

6) The effect of the mutation on OsNRAMP5 protein accumulation in yeast needs to be determined by western blot. Some of the effects may be related to decrease or increase in protein expression level. In the case of silent mutations (e.g. T507C), the codon usage bias could favor high expression in yeast. This further highlight the need to generate single point mutants of OsNRAMP5 using site-directed mutagenesis.

7) The discussion remains very superficial. As the 3D structure of NRAMP has been determined, the authors should show the residues they have identified on a 3D model of OsNRAMP5 rather than on the ball model presented in figure 6. Whether the residues affected by the mutations are conserved in other NRAMPs and/ or have been shown to be important for NRAMP structure or transport mechanism should be discussed. Probably, the authors should also mention the obvious perspective to introduce the mutations they have identified in rice using prime editing for example.

Author Response

Thank you for your comments to the content of the paper. And here are the responses to all the comments.

1) The introduction is solely focused on the mechanisms of Cd accumulation in rice and it lacks information on the mechanisms of metal transport by NRAMP proteins, structural information on NRAMP proteins and previous work on the structure-function relationships in this protein family. Especially, previous work already identified mutations that decrease Cd transport by other NRAMPs based on structural information (Bozzi et al., 2016) or on screen similar to the one that the authors report (Pottier et al., 2014).

Thank you for your kind comment. We have also looked through the papers related to the mechanism of NRAMP proteins, its structure and the relationship between transportation abilities and the protein structure. We have also added the additional information to the introduction and discussion part.

2) in figure 1, the authors show that, when expressed in yeast, OsNRAMP5 is able to complement the growth defect of the smf1 mutant on medium supplemented by EGTA and confers increased sensitivity to Cd. Both facts are firmly established in the literature and do not need to be illustrated again. In contrast, the authors do not show what happens in the conditions they have used for screening.

As the reader has to guess, the screen combines EGTA and Cd supplementation The authors have not considered the fact that EGTA binds Cd. Therefore, the free Cd concentration in the screening conditions is probably much lower than in the treatment with Cd only.

We agree that EGTA might also bind Cd, which may reduce Cd concentration in the screening conditions, and added an additional experiment with the medium containing 20 mM EGTA and 100 μM CdCl2 shown in Figure 1f. The result showed both of them had a limited growth compared to +Cd or +Mn environment respectively. Furthermore, in both Figure 2 and Figure 6, N5 showed a limited growth compared to M6 and M9, which indicated that the free Cd concentration was still enough to inhibit the growth of N5 even though some of them might be bound by EGTA. In this regard, the screening conditions should be appropriate to select plasmids with random mutations which are important for both Mn and Cd transportation.

2) Line 151-159: as mentioned above, the conditions of the screen and the library that was used for screening are not described. Based on the results shown in table 1, the same plasmids have been recovered many times. It is thus difficult to infer how many independent clones and mutations have been screened. Whether the screen was saturated is not indicated or discussed in the manuscript.

We feel sorry about the confusion on independence of clones. The random mutations were obtained from 10 PCR tubes separately, and they were ligated in the same vectors. Then the vectors with different insertions were transformed into the yeast and cultured on the separate plates with the same concentrations of Mn and Cd in medium. Under this situation, all the colonies were obtained from different plates, which means the library was saturated and the mutations recovered many times are likely to be important to transport Mn and Cd. We have clarified the procedures in the method part.

3) Line 163: the authors have not studied “Cd or Mn absorption” by the 20 mutants, as they claim. They have merely confirmed, looking at yeast growth, that they are still able to transport manganese and display lower sensitivity to Cd. This should be corrected.

Similarly, line 210, the authors have not measured any “metal-transport activity”, as they claim but again used yeast growth as a proxy to ability of the transport ability. This should be corrected.

Thank you for your comment on the studies on single metal absorption of either Mn or Cd. Since we have added an experiment on the solid medium with either EGTA or Cd with different concentrations, and in this experiment, mutants could always grow on the plates containing EGTA even at high concentration of EGTA that the growth of VC was inhibited. Meanwhile, mutants showed low sensitivity to Cd regardless of the Cd concentration, and they survived at medium and high Cd concentration where N5 had an inhibition in growth (Figure 1a-f). The results demonstrated mutants remained the ability of absorbing Mn as N5, but had a lower ability of absorbing Cd as VC, and the most significant difference would be found at 20 mM EGTA and 100 μM CdCl2, which was exactly the same result as Figure 5 showed. 

4) All the mutants identified have more than one mutation. To conclude about the effect of individual mutations, the authors have to generate single mutants using site-directed mutagenesis. This was done for the A512T (fig 5) but should also be done for the 2 other mutations reported.

We agree that the mutations at 8S and 111C may also be important for the metal absorption in OsNRAMP5. Meanwhile, the silent mutations at the 21st and 507th positions may contribute to the reason for the different absorption levels of metals comparing M6 and M9 as well, so we raised potential explanation in the discussion part and plan to generate single mutants at these positions in the future study. 

5) The authors have confirmed the effect of the mutations only on one concentration of Cd. To accurately determine the change in Cd sensitivity, different Cd concentrations should be tested.

Thank you for your suggestions on the experiment. We have added an additional experiment about the growth rate of yeast in 24 hours with different Cd concentration followed after Figure 3 in result part 2.3. Meanwhile, after M6 and M9 were selected from Figure 2, we added another experiment to test the Cd and EGTA sensitive of them in the solid medium with various Cd concentration, and the results showed the growth of mutants was not inhibited as N5 in all Cd conditions, but the growth of mutants was slightly worse than VC at high concentration of Cd (100 μM).

6) The effect of the mutation on OsNRAMP5 protein accumulation in yeast needs to be determined by western blot. Some of the effects may be related to decrease or increase in protein expression level. In the case of silent mutations (e.g. T507C), the codon usage bias could favor high expression in yeast. This further highlight the need to generate single point mutants of OsNRAMP5 using site-directed mutagenesis.

We agree that the silent mutants should also be investigated in both yeast and rice in future study because the protein expression level might vary and/or the improvement was specific in yeast protein, so we added an additional explanation in the discussion part to highlight the significance generation single point of OsNRAMP5.

7) The discussion remains very superficial. As the 3D structure of NRAMP has been determined, the authors should show the residues they have identified on a 3D model of OsNRAMP5 rather than on the ball model presented in figure 6. Whether the residues affected by the mutations are conserved in other NRAMPs and/ or have been shown to be important for NRAMP structure or transport mechanism should be discussed. 

Probably, the authors should also mention the obvious perspective to introduce the mutations they have identified in rice using prime editing for example.

We agree that 3D structure could show the locations of mutations more clearly, so 3D schematic diagrams of both before and after introducing mutation are added in addition to the ball model presented in Figure 7. Since the mutation at all three positions was not found in the previous study, it will be necessary to investigate whether such an effect is conserved in other NRAMPs as well. And the perspective of introducing the mutation in rice for a verification of the function of those amino acid is also mentioned in the discussion.

Reviewer 2 Report

Comments and Suggestions for Authors

This work generated random mutations in OsNRAMP5, and the authors used yeast to screen the key amino acid residue that effect Cd absorption, without interfering with the essential element Mn. The manuscript was well organized, but I still have some concerns:

1. Although it is a common sense that NRAMP5 is a transporter of both Mn and Cd. However, in Figure 1, we cannot directly draw such a conclusion based on the growth quality of the yeast. At least a concentration data in yeast is required.

2. I'm not very clear about the difference between pattern 1 and pattern 3 mutations.

3. The authors are advised to explain the reason for choosing M6 and M9 for subsequent growth and content determination. As can be seen from Figure 2b, perhaps M35, M36 and M70 are more necessary.

4. In figure 5, M6 exhibits increased concentrations of Mn and Zn, please give a reasonable explanation.

5. If possible, the authors are advised to compare the differences in the three-dimensional structures of proteins before and after the amino acid mutation.

Comments on the Quality of English Language

 Minor editing of English language required

Author Response

Thank you for your comments to the content of the paper. And here are the responses to all the comments.

1. Although it is a common sense that NRAMP5 is a transporter of both Mn and Cd. However, in Figure 1, we cannot directly draw such a conclusion based on the growth quality of the yeast. At least a concentration data in yeast is required.

Thank you for your comment. Although NRAMP5 is widely known as the transporter of both Mn and Cd, the experiment conditions which can clearly identify the difference in growth of yeast with or without OsNRAMP5 are quite different in different papers. In order to find the most appropriate conditions for our case, we checked the growth of both N5 and VC in different concentrations of EGTA and Cd (we have already updated the experiments with different concentration and added them into Figure 1), and found that with the environment of 20 mM EGTA chelating Mn or 100 μM CdCl2 (the actual result was shown in current Figure 1g). The clear difference in growth means that such conditions were appropriate to the later screening, so the same concentrations were applied in the screening. And we have already clarified the purpose of Figure 1 at the result section 2.1. 

2. I'm not very clear about the difference between pattern 1 and pattern 3 mutations.

We feel sorry for the confusion in the description of the difference of 4 patterns obtained in the screening. Both pattern 1 and pattern 3 shared the same mutation in amino acid, but pattern 1 had a silent mutation at the 21st position, and pattern 3 had a silent mutation at the 507th position. Similarly, both pattern 2 and pattern 3 shared the same mutations in amino acid, but pattern 2 had an extra silent mutation at the 291st position. We have added the clarification at the result section 2.2.

3. The authors are advised to explain the reason for choosing M6 and M9 for subsequent growth and content determination. As can be seen from Figure 2b, perhaps M35, M36 and M70 are more necessary.

We totally agree that M35, M36 and M70 from pattern 3 showed a better growth in Figure 2b. In this study, we only focused on the difference in amino acid and combined both pattern 1 and pattern 3 as the same type of mutation. Because the growth of yeasts and plants with the same gene introduced might be different depending on the characteristics of yeasts and plants, which means plants with pattern 3 plasmid introduced might not show as good performance as the yeast in this study.Meanwhile, since the number with pattern 1 was much larger than pattern 3, which indicated that pattern 1 might be a more common mutation, M6 from pattern 1 was chosen for the later experiments. We have explained the reasons for combining 4 patterns into 2 types, and the reason the yeasts were selected from each type at the result section 2.2 and 2.3.

4. In figure 5, M6 exhibits increased concentrations of Mn and Zn, please give a reasonable explanation.

Thank you for your clarification. Actually, we did not show the concentration of metals in original figure 5, but if you mean the original figure 4 (which became figure 5 now after experiments were added), we have just added an extra explanation on the increase of Mn and Zn of M6 under the environment without Cd in the discussion part about the silent mutations.

5. If possible, the authors are advised to compare the differences in the three-dimensional structures of proteins before and after the amino acid mutation.

We agree that 3D structure could show the locations of mutations more clearly, so 3D schematic diagrams of both before and after introducing mutation are added in addition to the ball model presented in Figure 7.

Round 2

Reviewer 2 Report

Comments and Suggestions for Authors

The authors have addressed my comments well.